# Research on Vibration Control Technology of Robot Motion Based on Magnetorheological Elastomer

**DOI:** 10.3390/ma15186479

**Published:** 2022-09-18

**Authors:** Xuegong Huang, Yutong Zhai, Guisong He

**Affiliations:** School of Mechanical Engineering, Nanjing University of Science & Technology, Nanjing 210094, China

**Keywords:** bipedal robot, dynamics simulation, MRE isolator, vibration reduction control algorithms

## Abstract

The vibration and impact of a humanoid bipedal robot during movements such as walking, running and jumping may cause potential damage to the robot’s mechanical joints and electrical systems. In this paper, a composite bidirectional vibration isolator based on magnetorheological elastomer (MRE) is designed for the cushioning and damping of a humanoid bipedal robot under foot contact forces. In addition, the vibration isolation performance of the vibration isolator was tested experimentally, and then, a vibration isolator dynamics model was developed. For the bipedal robot foot impact, based on the vibration isolator model, three vibration reduction control algorithms are simulated, and the results show that the vibration damping effect can reach 85%. Finally, the MRE vibration isolator hardware-in-the-loop-simulation experiment platform based on dSPACE has been built to verify the vibration reduction control effect of the fuzzy PID algorithm. The result shows the vibration amplitude attenuates significantly, and this verifies the effectiveness of the fuzzy PID damping control algorithm.

## 1. Introduction

Most current humanoid robots consist of rigid structures [1]. Although rigid structures obtain substantial strength and are also convenient to assemble, they result in robots that lack flexibility. During walking, running and jumping, the leg will inevitably collide with the ground at the moment of landing, and the contact reaction force from the ground will cause the whole leg and even the torso to be subjected to a certain degree of vibration from the bottom up [2]. Although the foot–ground collision is only momentary, the impact on the robot walking stability is significant. Impacts from the ground can cause sudden changes in the speed of the robot’s components, which can lead to problems such as tipping when the sudden change in speed reaches a certain value. It also interferes with the normal operation of the robot’s vision and sensing systems, causing serious damage to joints, servo motors and other structurally fragile areas and accelerating the destruction.

Therefore, the vibration damping of humanoid robots has become a hot research topic for scholars at home and abroad in recent years. This problem has been addressed by passive vibration isolation methods, such as installing cushioned support brackets at important parts [3] and adding impact-resistant materials to the bottom of the foot [4]. Arvind et al. at the MIT Bionic Robotics Laboratory designed a bionic robotic leg that mimics a musculoskeletal structure, using elastic bands to enhance flexibility [5]. Grizzle et al. at Michigan State University, USA, developed a bipedal robot that takes full advantage of the cushioning and energy-saving effects of springs at specific speeds and environments [6]. Youwang Yin et al. from Shanghai University of Technology proposed a new type of flexible vibration isolation structure, which achieves quasi-zero stiffness characteristics and achieves low-frequency vibration isolation with good effects [7].This type of method does not require external energy input and merely leverages the structural properties of the material itself to achieve damping, but it has a narrow operating band and is less adaptable [8]. Some scholars, at the same time, tried to isolate vibrations by active means. Bjorn Verrelst from the Free University of Brussels designed a bipedal robot with folded pneumatic muscles, and the corresponding active control algorithm controlled the pneumatic muscles to adjust the stiffness, showing an excellent cushioning and vibration damping effect [9]. Jonas Buchli et al. at the University of Southern California’s Computational Learning and Motion Control Laboratory utilized a quadruped robot developed by Boston Dynamics as a control object to achieve better robust robot motion in complex and rough terrain [10]. Shoukun Wang et al. from the Beijing Institute of Technology designed an adaptive impedance for wheeled robots based on the Lyapunov method to solve the robot’s vibration isolation problem with passing speed bumps [11]. Ji Sun et al. from Anhui University of Science and Technology designed a multidimensional vibration isolation device for aging robots based on the TRIZ theory, which provides an effective method for solving multidimensional vibration problems [12]. Ying Zhang et al. from Beijing Jiaotong University proposed a multidimensional vibration isolation platform for wheel-legged robotic vehicles, supplemented by spring damping modules, combining posture-adjusting and vibration control functions, with vibration isolation effects of over 50% [13]. Yunfeng Wang et al. combined the quasi-zero stiffness isolation system with active damping and investigated the effect of active damping and experimentally verified its good vibration isolation efficiency [14]. However, these methods all depend on continuous external energy input [15], which greatly increases the energy consumption problem of the robot, and the optimization algorithm needs to be improved for different robot structures to be applicable. In this context, the semi-active vibration isolation method is a technique that uses the controlled performance of intelligent materials to change the stiffness and damping of the structure in real time according to the external excitation to achieve vibration isolation, combining the advantages of active and passive vibration isolation. In recent years, many have attempted to apply the method to robot vibration damping problems, and it has become one of the main development directions for solving robot vibration damping problems.

Magnetorheological elastomers are prepared by mixing soft magnetic particles and polymer elastomers, which have the advantages of both liquids and elastomers, and overcoming the drawbacks of magnetorheological liquids, such as easy settling and poor stability. This incredible tunability makes magnetorheological elastomers a favorable application for semi-active vibration control in mechanical systems. Based on the superior stiffness damping controllability of magnetorheological elastomers, domestic and international scholars have made various attempts on magnetorheological elastomer vibration isolators for different practical engineering applications. Fengrong Bi et al. from Tianjin University designed a variable stiffness and damping damper based on MRE, which has obvious variable characteristics [16]. Fu et al. from Chongqing University developed a shear-compression composite mode MRE vibration isolator, demonstrating better performance in the MRE composite operating mode [17]. M.D. Christie at the University of Wollongong, Australia, developed an MRE vibration isolator for C-leg joints and measured that the robot leg can achieve up to 48% change in stiffness through a proprietary experimental platform built [18]. Weijia Ma et al. from Nanjing University of Science and Technology proposed an on–off control strategy for sweeping excitations and demonstrated good vibration isolation effects through experimental studies based on the frequency-shifting characteristics of magnetorheological elastomeric vibration isolator [19]. All of the above MRE isolators have excellent performance in transverse or longitudinal isolation or have been optimized for specific areas. It has also been demonstrated that great variable stiffness and variable damping characteristics can be achieved with composite directional vibration isolators.

The work proposed a composite bidirectional MRE vibration isolator designed for the vibration and impact problem of the robot after contact with the ground, using a magnetorheological elastomer, an intelligent material, to be installed on the foot of the robot. The stiffness and damping of the vibration isolators were adjusted in real time according to the vibration and shock they were subjected to, improving the humanoid robot’s impact resistance to complex environments. Performance tests, dynamics modelling and semi-active control of the MRE isolator were carried out in this paper. In order to simulate the actual situation of the robot walking, this paper completed a virtual prototype-based dynamics simulation to analyze the vibration and impact on the robot foot. It is adapted to the semi-active vibration isolator designed in this paper to provide a new solution to the robot vibration damping problem.

## 2. Simulation of Bipedal Robot Dynamics

The inverted pendulum model is a common model-based approach to gait planning for bipedal robots [20]. Bipedal robots generally have hip and knee joints, as well as ankle joints. In order to obtain the vibration and impact signals of the foot of the bipedal robot, it is modeled in a simplified way, and the gait planning of the bipedal robot is based on the inverted pendulum model to access the corner curves of its joints. The dynamics simulation in Adams was finished, following adding constraints, drives, sensors and other steps. The scaling parameters for the dynamics simulation are shown in Table 1. Setting the simulation time for the humanoid robot model, we can obtain the key frames for a single step cycle of the gait simulation, as shown in Figure 1 [21].

The data curves for the contact forces between the robot’s foot and the ground can be calculated through the Adams simulation analysis in Figure 2. It can be concluded from the figure that the humanoid robot has a periodic contact force with the ground during its movement and that the soles of its feet are mainly subjected to forces in the vertical direction. There are sharp peaks in the horizontal and vertical curves due to the gravity on the bipedal robot itself, relative speed of the foot to the ground and the friction between the foot and the ground when the robots first enter the unipedal support phase [22]. The walking dynamics simulation in this paper was carried out for a single step cycle, so during walking, the movement pattern of the left and right foot is symmetrical, and therefore, the curves of the left and right foot are similar in shape and have the same sharp peaks. Only the curves of the left and right foot have a phase difference in time. In the vertical direction, the contact reaction force on the left foot is the same as that on the right foot during the bipedal support period, while during the unipedal support period, the contact reaction force on the swing leg drops to zero, and the contact reaction force on the support leg, which bears the entire mass of the humanoid robot, is approximately doubled. The force on the foot in the horizontal direction may be related to frictional forces. In the simulation environment, there are vector marks representing the forces. These vector marks appear in approximately opposite directions. Set one of them in the positive direction and the other in the negative direction. The sharp peaks in Figure 2b are not strictly symmetrical, with some deviations owing to positive and negative directions artificially defined. Taking the foot–ground contact force as the input signal, we can get plantar vibration and impact signals, as shown in Figure 3.

## 3. Design and Experimentation of MRE Vibration Isolators for Robot Feet

### 3.1. Magnetorheological Elastomer-Based Vibration Isolator Design and Vibration Damping Performance Testing

According to the vibration situation of the robot’s feet, a vibration isolator based on the magnetorheological elastomer is designed. Given the relationship between the direction of motion of the magnetorheological elastomer in the vibration isolation support and the direction of the applied magnetic field, the two main modes of operation are the shear mode and pull–press mode, the principle of which is shown in Figure 4. As relevant studies show, the modulus of the magnetorheological elastomers in the pull–compression mode in the zero-field case is twice that in the shear mode [23,24], while the amount of change in the compressive and shear module is essentially the same for the same magnetic field strength. The MRE in shear mode of operation allows for greater magnetorheological effects. Meanwhile, the MRE in the pull–pressure mode has a greater load-bearing capacity.

In this manuscript, carbonyl iron powder (type: MPS-MRF-15) is chosen as the magnetic particle. Silicone rubber (type: RTV-704) with good adhesion and aging resistance is chosen as the elastic matrix, and dimethyl silicone oil is chosen as the auxiliary additive. The carbonyl iron powder, dimethyl silicone oil and 704 silicone rubber were first mixed in a beaker at a mass ratio of 8:1:1 and dispersed well using a mixer. The mixture was then placed in a vacuum drying oven to extract the air inside from the stirring process in three stages, followed by pouring the defoamed mixture into a mold and placing it in the drying oven again to extract the air three times and, finally, covering the mold with a lid and placing it in a magnetizing device at 800 mT for 2 h to pre-structure the anisotropic MRE material.

The force on the foot during robot walking is a multidirectional force, which inevitably leads to vibration and shock in multiple directions. Conventional MRE isolators for one direction only are not a reasonable solution to this problem. This manuscript therefore attempts to design a composite bidirectional MRE isolator, which is capable of isolating both vertical and horizontal directions. The working principle of the designed composite bidirectional MRE isolator is expressed in Figure 5. The vertical vibration isolation of this composite bidirectional MRE is accomplished by a hollow circular MRE and a disc-shaped MRE (i.e., disc MRE1). A physical diagram of the MRE material is shown in Figure 6. When the vibration isolator is subjected to forces in the vertical direction, the disc MRE1 will operate in the pull-press mode, which gives the vibration isolator a greater load carrying capacity in the vertical direction, while the circular MRE, which is fixed between the upper and lower coils by hexagonal fastening screws, will operate in the shear mode in a similar way, which has a greater magnetorheological effect than the disc MRE1. In other words, the range of variation in stiffness and damping will be more notable. These two magnetorheological elastomers enable the vibration isolator to possess a large load-bearing capacity in the main vibration isolation direction and a large range of stiffness and damping variation, making it well-suited to large load-bearing and large impact conditions. Meanwhile, the horizontal vibration isolation, which is relatively weak in terms of vibration and shock, is completed by another disc-shaped MRE (i.e., disc MRE2). Its operation in shear mode allows it to solve the problem of vibration isolation in the horizontal direction to a certain extent. It is worth noting that MRE1 can hardly isolate horizontal vibrations, such as MRE2.

The bidirectional MRE isolator designed in this paper is a strictly cylindrical axisymmetric structure, so the simulation analysis is carried out by a one-half model in the X-Z plane. The material properties of each part are first defined via Maxwell’s material library, followed by the selection of the appropriate air boundaries as boundary conditions. Then, we define the loading current density and the number of winding turns for the coil. Finally, the division of the mesh and the solution options setting are made, and then, the simulation can be carried out. The magnetic field distribution of the MRE vibration isolator the one-half model with a 2-A current applied is shown in Figure 7. In Figure 7, the magnetic circuit in the MREs meets the design expectations. In the one-half simulation diagram, the MREs are all represented by black rectangles. The center of the rectangles is selected as the location where the magnetic induction intensity is recorded, varying the current (0–2 A) for multiple simulations. According to the recording, the simulations in Maxwell-2D give the relationship between the coil current and the magnetic induction strength in the three MREs, as shown in Figure 8.

MREs are a type of viscoelastic material, which means both viscous and elastic deformation mechanisms exist. The mechanical properties of viscoelastic materials are shown in Figure 9 [25].

Based on the mechanical properties, the following equations for equivalent stiffness and equivalent damping are deduced [26]:(1)$$K_{eff}=\frac{F_{xmax}-F_{xmin}}{X_{max}-X_{min}}$$
(2)$$C_{eq}=\frac{S}{2\pi^{2}fX^{2}}$$
where *K_eff_* denotes the equivalent stiffness of the isolator, *X_max_* and *X_min_* denote the maximum and minimum displacement values of the MRE isolator and *F_xmax_* and *F_xmin_* denote the force output of the MRE isolator at the maximum and minimum displacement values, respectively. *C_eq_* denotes the equivalent damping of the isolator, *S* denotes the area of the force–displacement hysteresis loop, *f* denotes the excitation frequency and *X* denotes the excitation amplitude.

In order to further establish the mechanical performance of the designed MRE vibration isolator, a performance test rig was set up for testing in both the vertical and horizontal directions. The sinusoidal excitation was used to perform the experiments in order to verify the vibration isolator’ s damping and vibration isolation performance. Simulation experiments for random signals will be carried out in our future research. In addition, both the sinusoidal excitation and the foot–ground contact force are periodic signals. Therefore, experiments with periodic signals are more in line with the scenario of robots. During the test experiments in the vertical direction, the signal generator was to generate different sinusoidal excitations, while the current provided by the DC supply was increased from 0 A to 2 A. The slope of the long axis of the ellipse represents the equivalent stiffness of the isolator, and the area of the ellipse represents the equivalent damping of the isolator as the applied current increases, as shown in Figure 10 for the vertical direction. The equivalent stiffness and damping of the MRE isolator are therefore positively related to the applied current.

In the horizontal direction, aiming to test the performance of the MRE vibration isolator under more working conditions, the signal generator was to generate sinusoidal excitation frequency of 8 Hz, and the displacement amplitudes at each excitation frequency were 0.4 mm, 0.6 mm and 0.8 mm, respectively. The current supplied by the DC supply is from 0 A to 2 A. The 8-Hz sinusoidal excitation experiments are shown in Figure 11. Compared to the force–displacement hysteresis loop in the vertical direction, the overall trend is the same, with the equivalent stiffness and damping of the MRE isolator being positively correlated with the applied current, and the higher the current, the higher the peak under the same conditions. In a word, the results of this experiment are consistent with the theory. In the previous section on the design principles of the vibration isolator, a total of three MREs operate in the vibration isolator. The circular MRE and MRE1 are responsible for causing the stiffness and damping changes in the axial direction (vertical direction), and MRE2 is responsible for causing the stiffness and damping changes in the radial direction (horizontal direction). Therefore, there are two MREs operating in the axial direction and only one MRE in the radial direction. That is why the theoretical dependence of the axial direction on the current is more significant. This explains the much less current dependence in Figure 11. However, as can be seen from the two figures, the trend in the hysteresis loop is similar.

Liu Tao et al. [27,28] proposed a shear mode MRE isolator. The isolator is applied to a unidirectional seismic isolation system and has almost no variable stiffness characteristics in the vertical direction of the isolation direction, although it has a good dynamic response in the isolation direction. Compared to that isolator, our isolator has a dynamic response in both directions and can be controlled by the appropriate algorithm to achieve variable stiffness and variable damping in both directions. In terms of the vibration isolation direction, the originality is conspicuous.

### 3.2. Dynamical Modeling of the Magnetorheological Elastomeric Vibration Isolator

Both simulation and semi-physical experiments require a dynamic model of the vibration isolator. In this paper, the Kelvin model is selected to describe the mechanical properties of the MRE vibration isolator, which mainly equates the viscoelastic material as a combination of spring and viscous pot units. In conjunction with the foot–ground collision in the previous gait simulation of robot walking, the MRE vibration isolation system is modeled as shown in Figure 12.

From the dynamics model, the equations of motion for this MRE vibration isolation system can be obtained as follows:(3)$$\left\{ \begin{array}{l} m_{1}{\ddot{x}}_{1}=k_{1}\left(x_{0}-x_{1}\right)+c_{1}\left({\dot{x}}_{0}-{\dot{x}}_{1}\right)+F_{c}\\ m_{0}{\ddot{x}}_{0}=F-F_{c}-k_{0}x_{0}-c_{0}{\dot{x}}_{0}-k_{1}\left(x_{0}-x_{1}\right)-c_{1}\left({\dot{x}}_{0}-{\dot{x}}_{1}\right)\\ F_{c}=\Delta k\left(x_{0}-x_{1}\right)+\Delta c\left({\dot{x}}_{0}-{\dot{x}}_{1}\right) \end{array}\right.$$
where *m*_0_ denotes the unsprung mass, i.e., the part of the robot below the MRE isolator, and *m*_1_ denotes the sprung mass, i.e., the sum of the mass of the robot body and the mass of the core in the MRE isolator. *k*_0_ and *c*_0_ denote the stiffness and damping coefficients set during the collision between the robot’s foot and the ground in the gait simulation, respectively. *k*_1_ and *c*_1_ denote the initial equivalent stiffness and damping of the vibration isolator without an applied magnetic field. Δk and Δc denote the equivalent stiffness and damping of the vibration isolator as the magnetic field changes after a magnetic field is applied. *x*_0_*(t)* denotes the excitation displacement, i.e., the vibration displacement of the robot after vibration and shock, and *x*_0_*(t)* denotes the response displacement, i.e., the vibration displacement transmitted to the robot body through the MRE isolator. *F* denotes the foot–ground contact force obtained in the previous simulation, and *F_c_* denotes the magnetic field control force generated when the isolator is energized, which is obtained by controlling the equivalent stiffness and damping of the MRE isolator.

## 4. Simulation Experiments of Control Algorithms Based on Foot–Ground Signals

The vibration isolator model is known to be in contact with the robot’s foot–ground signal. Simulated vibration isolation experiments are accomplished in Simulink for three algorithms and compared to the vibration isolator output for the no-isolation case.

### 4.1. Fuzzy Control

Fuzzy control is robust and well-suited to the control study of MRE vibration isolators. The design flow of the fuzzy control focuses on the design of its controller, which is built in Simulink in a block diagram, as shown in Figure 13. The input variables are divided into seven classes: NB, NM, NS, ZO, PS, PM and PB, while the applied current as an output variable is divided into only four fuzzy classes: ZO, PS, PM and PB, as there are no negative values. The input affiliation function is of Gaussmf type, with the domain set to [−2, 2], and the output affiliation function is of trimf type, with the domain set to [0, 2]. As both input variables have seven fuzzy levels, the fuzzy rules should be designed as 49 levels. The design process is based on the corresponding theoretical relationships analyzed from the experience and experimental results of various scholars in the field of vibration control [29].

### 4.2. PID Control

The PID control law adopted for the vibration isolators in this paper is as follows:(4)$$u=k_{P}\left[e\left(t\right)+\frac{1}{T_{I}}\int_{0}^{t}e\left(t\right)+\frac{T_{D}de\left(t\right)}{dt}\right]$$
where *e(t)* denotes the deviation, i.e., the difference between the actual output and the theoretical value. *k_p_*, *k_i_* and *k_d_* denote the scale factor, the integral time constant and the differential time constant, respectively. The magnetically induced control force of the MRE isolator cannot meet the control force output by the PID controller at all times, so the control force output by the PID must be limited, as shown in Equation (5):(5)$$u=\left\{ \begin{array}{ll} 0 & u\cdot F\leq 0\\ u & u\cdot F>0\;\&\;abs\left(u\right)\leq abs\left(F_{\max}\right)\\ F_{\max} & u\cdot F>0\;\&\;abs\left(u\right)>abs\left(F_{\max}\right) \end{array}\right.$$
where *u* denotes the control force output from the PID controller, and *F_max_* denotes the maximum magnetically induced control force of the vibration isolator.

### 4.3. Fuzzy PID Control

The fuzzy PID control algorithm adjusts the parameters of the PID in real time through the inference capability of the fuzzy controller. Fuzzy PID control algorithms come in various forms, but their basic operating principle remains the same. The fuzzy PID algorithm uses deviation *e* and the deviation change rate *de/dt* as the input of the fuzzy controller, output *k_p_*, *k_i_* and *k_d_* and reasoning using fuzzy control rules to rectify the three parameters of the PID in real time, and eventually, the PID controller completes the real-time control of the controlled object. The setup of this control algorithm is similar in principle to the first two, where the fuzzy inference rules draw on the practical experience of relevant experts, and the rules for the three parameters are similar. A block diagram is therefore established in Simulink as follows in Figure 14 [30]:

### 4.4. Semi-Active Control Simulation Results and Analysis

Based on the foot contact force curve and the MRE vibration isolator dynamics model, a block diagram of the integrated MRE vibration isolation control system was constructed, as shown in Figure 15. The results after the four control methods are shown in Figure 16, and their specific details are shown in Table 2.

The analysis of the results in conjunction with the above leads to the following conclusions:

(1) Overall, the MRE vibration isolator seems to have a fabulous damping effect in the robot walking process. The peak acceleration and root mean square (RMS) values of the robot foot vibration signal are reduced by more than 85% under the MRE vibration isolator’s processing. The vibration displacement amplitude is within 4 mm, which means it has a small impact on the robot control accuracy.

(2) From the acceleration control effect, comparing several control algorithms, PID control reveals the best control effect on the acceleration peak, followed by fuzzy PID control, while fuzzy control has the third-place effect. The acceleration RMS value is the best effect of fuzzy PID control algorithm, followed by PID control, and the third-place is still fuzzy control.

(3) In terms of displacement control, the PID control algorithm is the least effective, the fuzzy control is the second-most effective, and the fuzzy PID has the highest peak displacement. This is because, in buffered damping, in order to cope with the impact of higher acceleration, a certain amount of vibration amplitude is demanded to better offset the impact effect, i.e., by increasing the displacement in exchange for a tinier acceleration. Therefore, fuzzy and PID control of the displacement control is more expressive than fuzzy PID.

## 5. Hardware-in-the-Loop-Simulation of Vibration Isolator Damping Control Based on Foot–Ground Signal

To investigate the rationality of the control algorithm, a test bench for vibration isolation systems based on dSPACE is built, as shown in Figure 17. dSPACE is a semi-physical real-time simulation system that can directly interact with MATLAB/Simulink modules. The software part is mainly responsible for the controller design, automatic code generation and download, including the visualization and management of experiments; the hardware part mainly consists of a high-performance processor board, a multi-channel high-speed A/D converter board and a high-resolution D/A converter board.

The experimental procedure for the whole vibration isolation control system is as follows: Firstly, the control system is set up in a computer using the MATLAB/Simulink R2018a module, as shown in Figure 18. The signal generator emits a sinusoidal waveform with a frequency corresponding to the foot–ground contact, which is amplified by a power amplifier to drive the exciter for excitation. The input acceleration to the controller is obtained by collecting the signal from sensor 1, while the signal from sensor 2 is collected by another ADC module, which is connected to a charge amplifier for conversion and then connected to the board for acquisition. The current–voltage conversion is based on the coil resistance of the MRE isolator. The output of the board is connected to the winding coil of the MRE isolator to complete the control of the applied current. The software module completes the code-related task, and the simulation results are obtained in controldesk, as shown in Figure 19.

Figure 18 shows the acceleration response curve obtained by the experiment. As can be seen from the figure, the response acceleration amplitude decreases to a certain extent at 10 s after the fuzzy PID controller control, and from the overall peak value, the acceleration peak value decreases from 0.5736 m/s^2^ without the control to 0.3539 m/s^2^ with the fuzzy PID control applied, which decreases by 38.3% and has a better control effect. The effectiveness and reasonableness of the fuzzy PID algorithm were confirmed.

## 6. Conclusions

This paper presents a composite bidirectional MRE vibration isolator solution for the cushioning and vibration damping of a robot foot: a composite bidirectional MRE vibration isolator is designed using magnetorheological elastomers with high magnetorheological effects. Based on the vibration and impact signals to which the foot is subjected during walking, three controllers: fuzzy, PID and fuzzy PID, are designed, and the damping effects of the control algorithms are compared with a simulation analysis. The experimental platform of the vibration isolation control system is finally built by means of a dSPACE hardware-in-the-loop-simulation system, and the effectiveness of the MRE vibration isolator and the designed control algorithm is verified for sinusoidal excitation signals. The significances and impacts of this work are as follows:

Firstly, this manuscript proposes a dynamical approach to the gait analysis of a bipedal robot, providing an important vibration isolation perspective where the forces on the robot’s foot cause vibrations. Secondly, this paper designed a bidirectional vibration isolator using MREs with high magnetorheological effects and conducted experiments to demonstrate its good bidirectional dynamic performance, offering a solution for scenarios requiring multidirectional engineering vibration isolation. Finally, three intelligent control algorithms with good results were designed in this paper, enriching the ideas of intelligent algorithm control.

## Figures and Tables

**Figure 1 materials-15-06479-f001:**
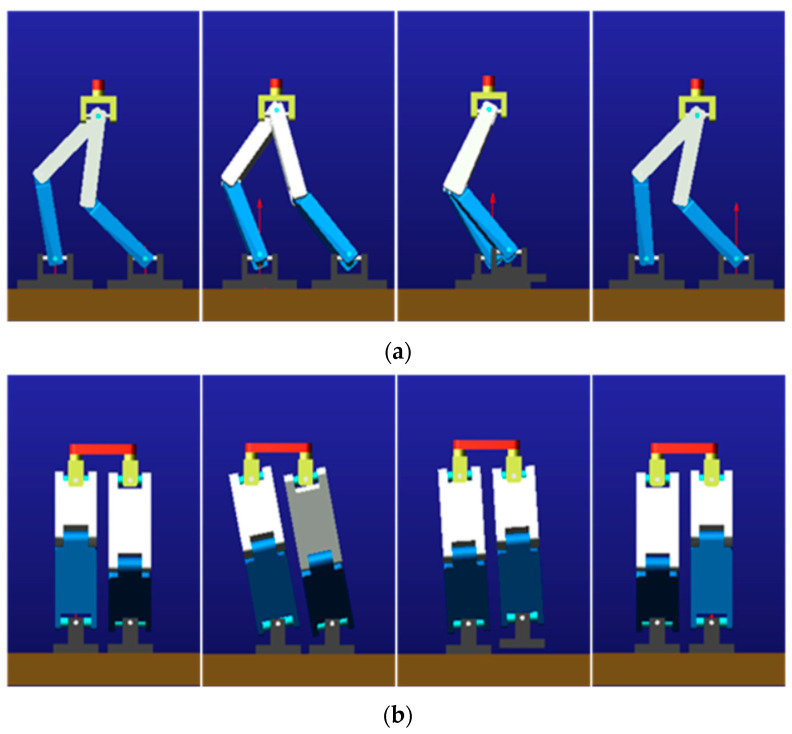
Kinetic simulation keyframes: (**a**) forward plane and (**b**) lateral plane.

**Figure 2 materials-15-06479-f002:**
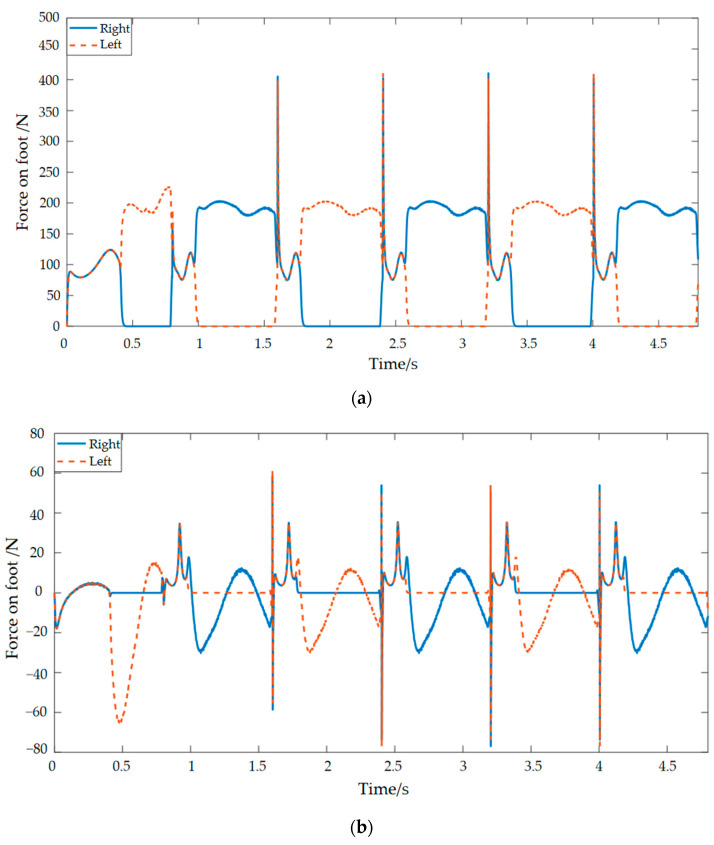
Humanoid robot with ground contact forces: (**a**) vertical direction and (**b**) horizontal direction.

**Figure 3 materials-15-06479-f003:**
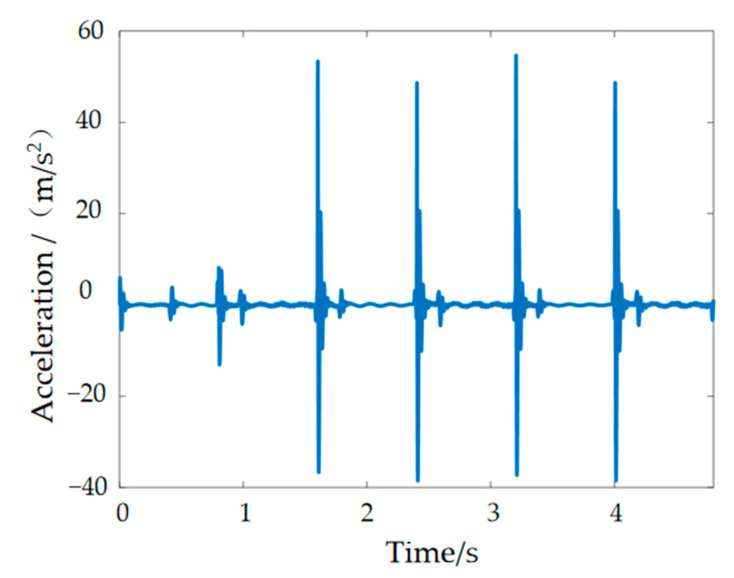
Plantar vibration and impact signals.

**Figure 4 materials-15-06479-f004:**
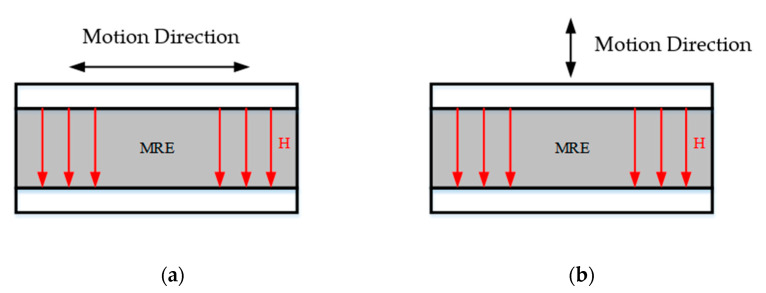
Operating modes of the magnetorheological elastomers: (**a**) shear mode and (**b**) pull–press mode.

**Figure 5 materials-15-06479-f005:**
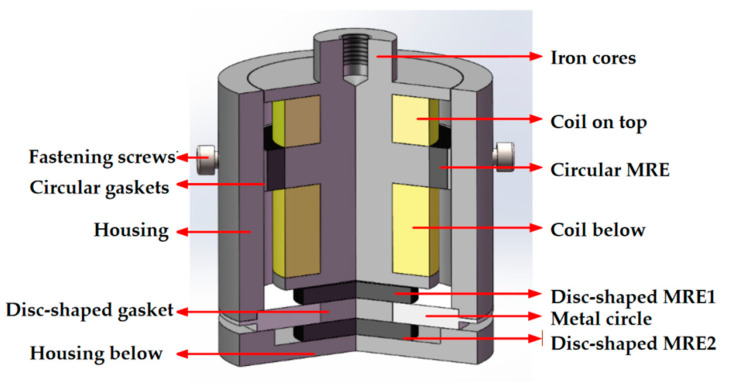
Composite bidirectional MRE vibration isolators.

**Figure 6 materials-15-06479-f006:**
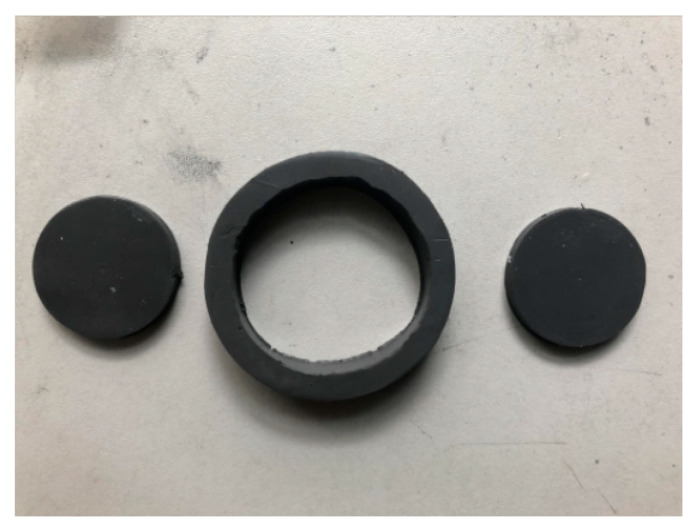
Physical view of the MRE.

**Figure 7 materials-15-06479-f007:**
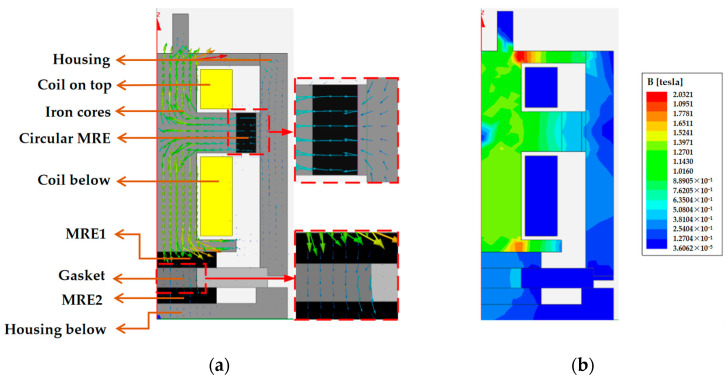
Simulation of the magnetic field of a vibration isolator under a 2-A applied current: (**a**) vector image and (**b**) magnitude image.

**Figure 8 materials-15-06479-f008:**
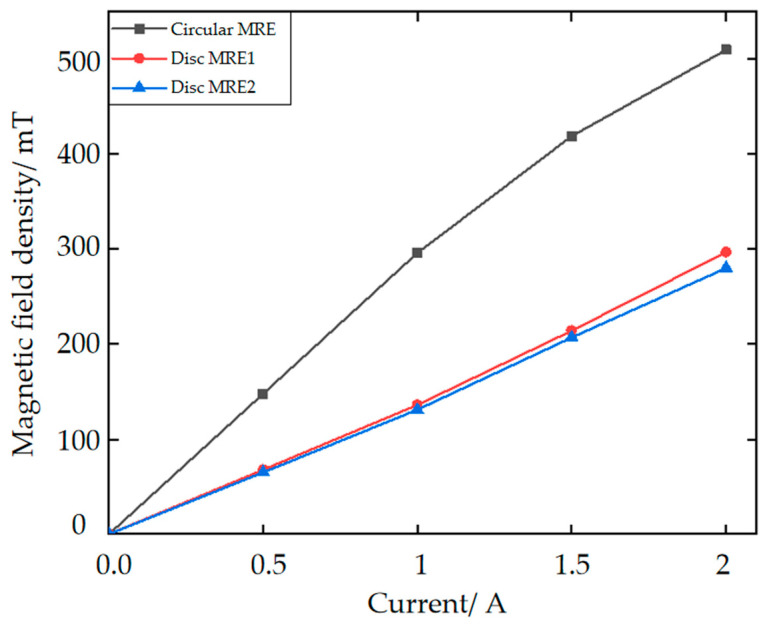
Trend of the magnetic induction strength with the current.

**Figure 9 materials-15-06479-f009:**
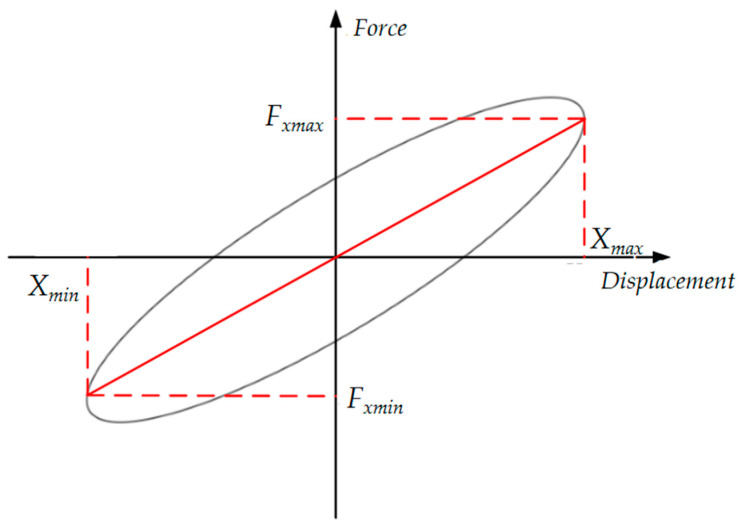
Force–displacement hysteresis loops.

**Figure 10 materials-15-06479-f010:**
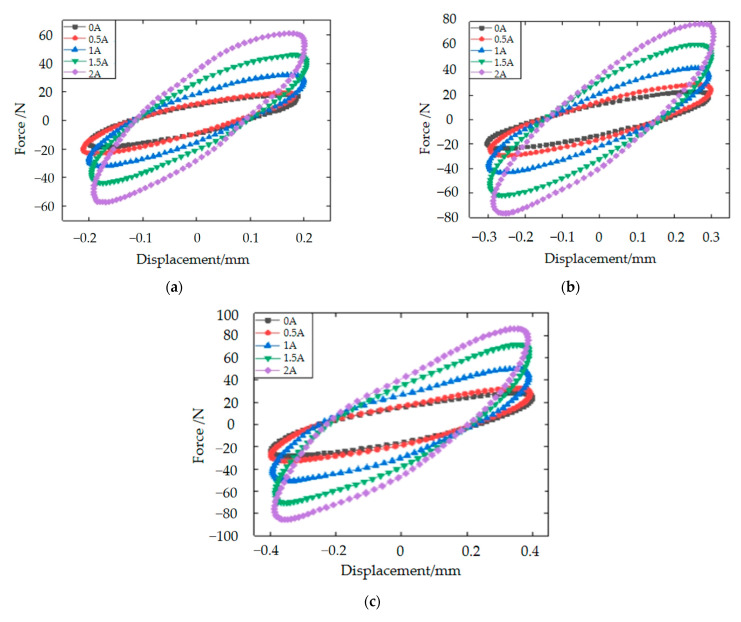
Force–displacement hysteresis loops in the vertical direction of the MRE isolator for 8-Hz excitation: (**a**) 8 Hz, 0.2 mm; (**b**) 8 Hz, 0.3 mm and (**c**) 8 Hz, 0.4 mm.

**Figure 11 materials-15-06479-f011:**
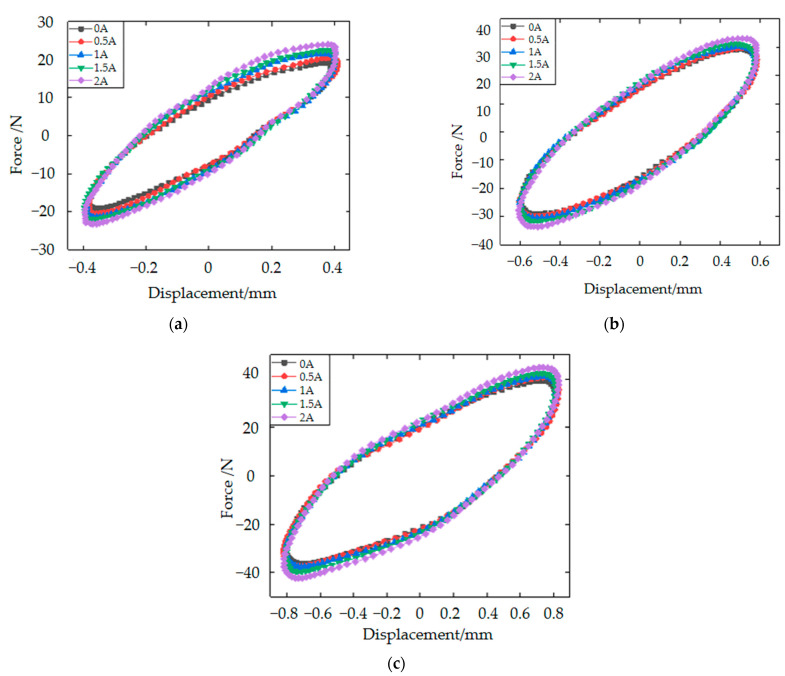
Force–displacement hysteresis loops in the horizontal direction of the MRE isolator for 8-Hz excitation: (**a**) 8 Hz, 0.4 mm; (**b**) 8 Hz, 0.6 mm and (**c**) 8 Hz, 0.8 mm.

**Figure 12 materials-15-06479-f012:**
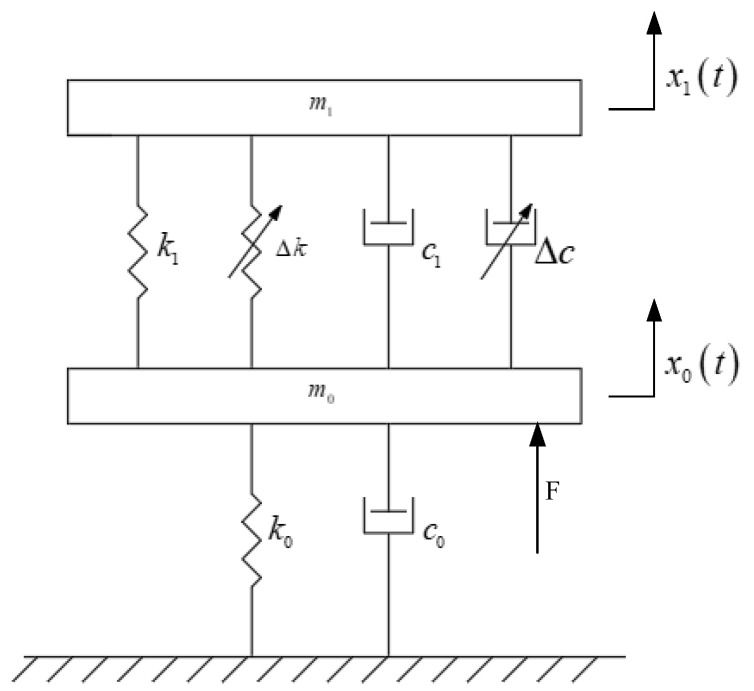
Dynamical model of the MRE vibration isolation system.

**Figure 13 materials-15-06479-f013:**
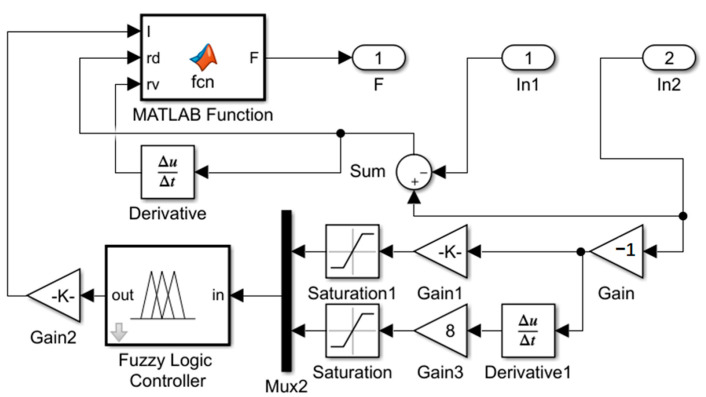
Fuzzy controller block diagram.

**Figure 14 materials-15-06479-f014:**
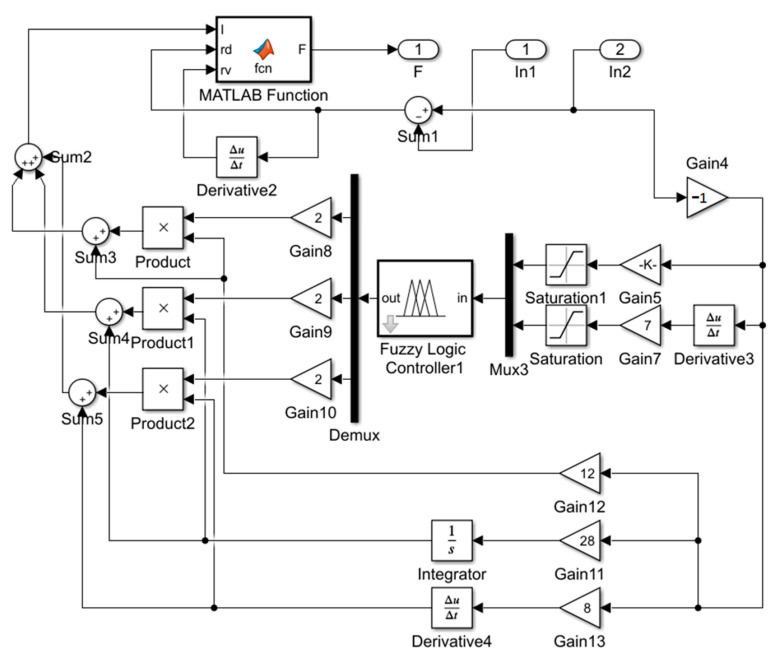
Fuzzy PID controller block diagram.

**Figure 15 materials-15-06479-f015:**
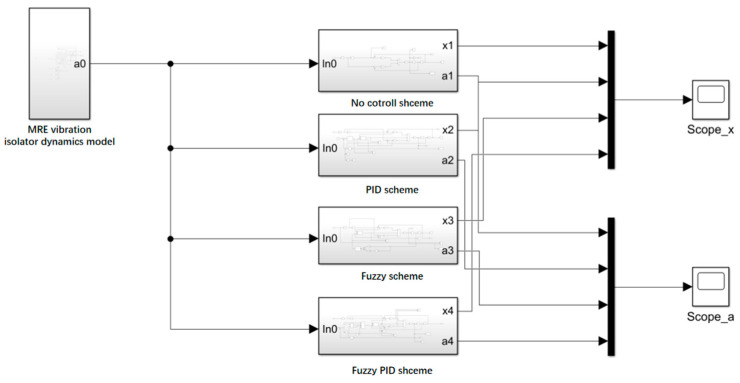
Block diagram of the MRE vibration isolation control system.

**Figure 16 materials-15-06479-f016:**
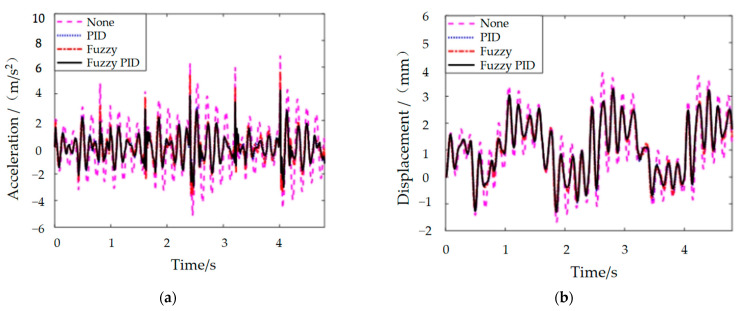
MRE vibration isolator response signal: (**a**) acceleration and (**b**) displacement;.

**Figure 17 materials-15-06479-f017:**
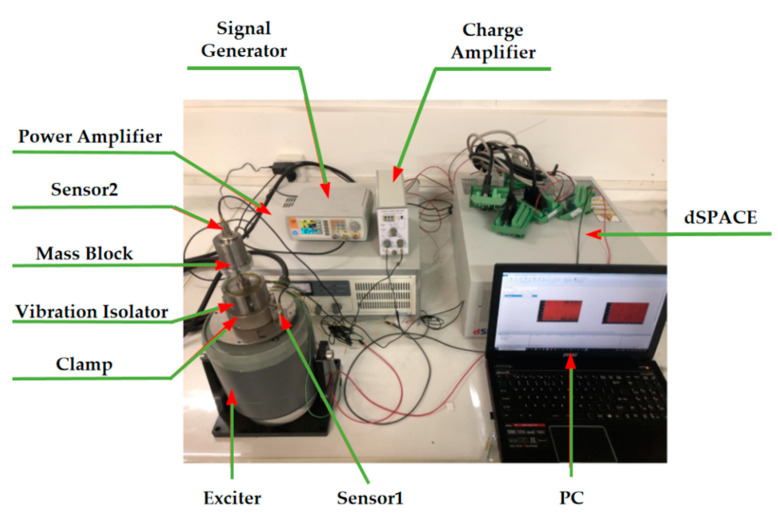
Experimental platform.

**Figure 18 materials-15-06479-f018:**
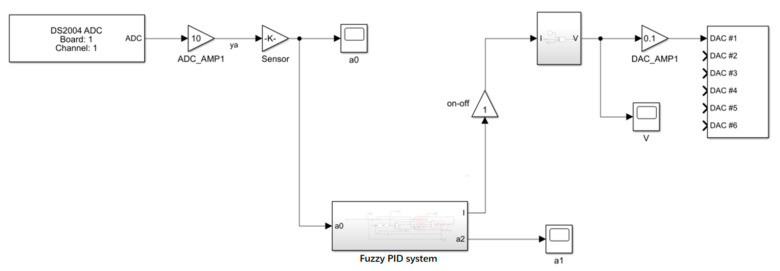
Block diagram of the control system.

**Figure 19 materials-15-06479-f019:**
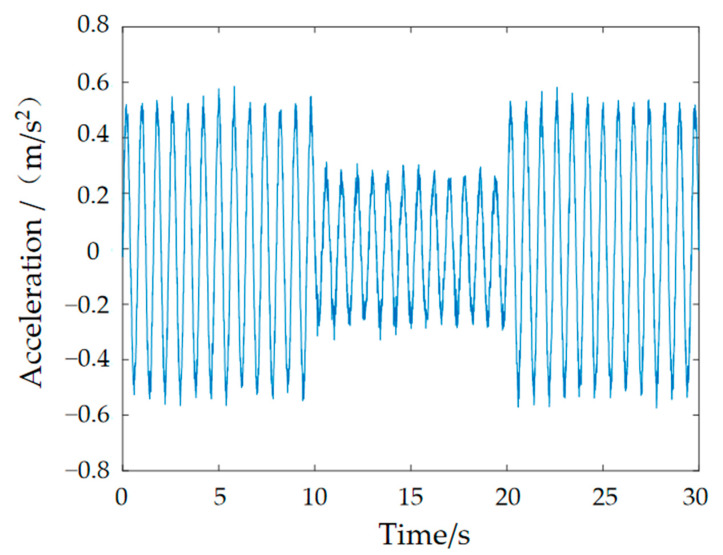
Response acceleration curve.

**Table 1 materials-15-06479-t001:** Kinetic scaling parameters in Adams.

Parameters	Values and Units
Stiffness factor	10^5^ (N/mm)
Force Index	2.2
Damping factor	10 (Ns/mm)
Penetration depth	0.1
Coefficient of static friction	0.5
Coefficient of dynamic friction	0.3
Size of each foot	10 × 5 × 3 cm^3^
Mass of each foot	100 g
Simulation time	9.6 s

**Table 2 materials-15-06479-t002:** Response of the vibration isolators under vibration signals.

	None		PID		Fuzzy		Fuzzy PID	
	Peak	RMS	Peak	RMS	Peak	RMS	Peak	RMS
displacement	3.871	1.604	3.214	1.4452	3.259	1.4398	3.289	1.4536
acceleration	6.872	1.978	4.266	1.0076	5.691	1.0623	4.267	0.997

## Data Availability

The data used to support the findings of this study are available from the corresponding author upon request.

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
