# Peer review of "Research on Vibration Control Technology of Robot Motion Based on Magnetorheological Elastomer"

_materials, 2022, doi:10.3390/ma15186479_

Round 1
Reviewer 1 Report
The document requires an English reviews, there are several typos and sentences that are incoherent, e.g. at line 29 “Robot as a typical unstable system, although the foot-ground collision is only momentary, the impact on the robot walking stability is significant”, what do the authors mean? Line 34 “joint joint”
The originality of the proposed vibration isolator is not clear, there are several works related to such designs, and the experimental results need to be compared to other techniques to conclude advantages.
At Figure 2 the units (m/s^2) correspond to acceleration, not to force.
With respect to the compared control schemes, on the one hand in eq. (5) line 266, the notation “u(t). Fc” causes confusion, it can be interpreted as product instead of a clause of selection; and on the other hand, there is a lot of missing information related to the fuzzy control, in order to reproduced the results, fuzzy sets, etc.
Plots in figure 15 cannot be distinguished.
Reviewer 2 Report
The model is not well represented in the experimental work. What were the scaling parameters for the model foot? The scaling parameters include length, mass, acceleration, and time.
The analyses were conducted based on sinusoidal excitations. In real life, the excitations are random. Why did the authors not choose to carry out control simulations with random excitations?
There are not sufficient details about the fabrication of the MREs. The fabrication methodology must be detailed. What were the Wt% of iron particles? What were the base materials? How did you fabricate the samples?
There are not sufficient discussions about the magnetic field simulations. Detail the simulation methodology. Show the magnetic field distribution on the MREs in different axes. The magnetic field intensities were shown in Figure 7 for different current inputs. But at what location on the MREs?
Check Figures 2a and b. What does the y-axis represent? Force does not have the unit of m/s2.
Are Figures 9 and 10 supposed to be for both the vertical directions?
The introduction section must be expanded. Include more references and preferably more recent publications.
Reviewer 3 Report
This paper presents a composite bidirectional MRE vibration isolator solution for cushioning and vibration damping of a robot foot. The research was proper designed and of good scientific soundness. I would recommend it for publication in Materials if my following comments can be addressed:
1. In page 1 line 6, why the author’s name abbreviations are D. Y. and S. W.?
2. More detailed description of the ground contact force simulation results (figure 2) are required. What's the explanation those sharp peaks, and why they those peaks exist simultaneously on both left and right foot? What does positive and negative sign mean in figure 2b? Why the sharp peaks in figure 2b are asymmetric on positive and negative directions?
3. In the structure shown in figure 5, what's the difference between MRE1 and MRE2? Can MRE1 also isolate horizontal vibrations like MRE2?
4. When describing the force-displacement hysteresis loop in horizontal directions, there are many mismatches between the text and figure 10. In page 9 line 205-207, the authors said "frequencies of 4Hz, 6Hz and 8Hz, and the displacement amplitudes at each excitation frequency were 0.4mm, 0.6mm and 0.8mm respectively. The current supplied by the DC supply is constant." However, in figure 10, the captions show "vertical directions", "(a) 8Hz,0.2mm; (b) 8Hz,0.3mm;(c)8Hz,0.4mm", and the DC current varies from 0 to 2A. Please correct them.
5. In page 8 line 208, the authors concluded "the overall 208 trend is the same", but the results in figure 9 and figure 10 show obvious differences. Please address and explain the different behavior between figure 9 and 10. Why figure 10 shows much less current dependence.
6. In the conclusion part, please address what are the significances and impacts of this work.
7. Please thoroughly check your language. So many grammar mistakes are found throughout the manuscript. E.g., page 1 line 34: "joint joints"; page 1 line 35: "accelerating the speed of"; page 14 line 345...
Reviewer 4 Report
1. This paper presents a composite bidirectional MRE vibration isolator solution for cushioning and vibration damping of a robot foot: a composite bidirectional MRE vibration isolator is designed using magnetorheological elastomers with high magnetorheological effects. Based on the vibration and impact signals to which the foot is subjected during walking, three controllers, fuzzy, PID and fuzzy PID, are designed and the damping effects of the control algorithms are compared with simulation analysis. The MRE vibration isolator hardware-in-the-loop-simulation experiment platform based on dSPACE has been built to verify the vibration reduction control effect of fuzzy PID algorithm. Overall, the paper structure is complete. The picture is clear and readable. The discussion of the results and the conclusion allow relevant knowledge to be drawn.
2. It is recommended to add Journal articles in the last three years.
Round 2
Reviewer 1 Report
The new version has greatly improved, the experiments and simulation cases are well explained, and the the introduction presents new references.
A deep review of english is highly suggested, teher are still a lot of english mistakes, and mainly in the green highligthed text.
Reviewer 2 Report
Label the main components in Figure 7.
Other than that it looks like the authors answered the Reviewer's comments and concern.
